# Dynamic topic modeling of twitter data during the COVID-19 pandemic

**Alexander Bogdanowicz**[1][�} , **ChengHe Guan**[1,2][�} *

**1** New York University Shanghai, Shanghai, China, **2** Shanghai Key Laboratory of Urban Design and Urban Science, NYU Shanghai, Shanghai, China

} These authors contributed equally to this work.

\* chenghe.guan@nyu.edu

## Abstract

In an effort to gauge the global pandemic's impact on social thoughts and behavior, it is important to answer the following questions: (1) What kinds of topics are individuals and groups vocalizing in relation to the pandemic? (2) Are there any noticeable topic trends and if so how do these topics change over time and in response to major events? In this paper, through the advanced Sequential Latent Dirichlet Allocation model, we identified twelve of the most popular topics present in a Twitter dataset collected over the period spanning April 3rd to April 13th, 2020 in the United States and discussed their growth and changes over time. These topics were both robust, in that they covered specific domains, not simply events, and dynamic, in that they were able to change over time in response to rising trends in our dataset. They spanned politics, healthcare, community, and the economy, and experienced macro-level growth over time, while also exhibiting micro-level changes in topic composition. Our approach differentiated itself in both scale and scope to study the emerging topics concerning COVID-19 at a scale that few works have been able to achieve. We contributed to the cross-sectional field of urban studies and big data. Whereas we are optimistic towards the future, we also understand that this is an unprecedented time that will have lasting impacts on individuals and society at large, impacting not only the economy or geo-politics, but human behavior and psychology. Therefore, in more ways than one, this research is just beginning to scratch the surface of what will be a concerted research effort into studying the history and repercussions of COVID-19.

## Introduction

The last two decades have seen societies continue to evolve their means of virtual socializing and self-expression. Consequently, simultaneous advancements in the primary statistical domains related to communication via social media (i.e. Natural Language Processing (NLP)) are observed [1–3]. With the advent of a global pandemic impacting the lives of billions across the globe, the capability to assess how individuals, groups, and societies respond to, and cope with, the extraordinary consequences of the COVID-19 pandemic is ever more pressing. In as early as late November of 2019, a SARS-like virus began spreading between neighborhoods in

**Data Availability Statement:** According to Twitter's Terms of Service, we may only distribute Tweet IDs, Direct Message IDs, or User IDs. This means that we're not able to share datasets containing the content of tweets. We stripped the datasets and provided a list of Tweet IDs, please

refer to the file: twitter_data_tweet_ids https://doi.org/10.7910/DVN/YXIAEK.

**Funding:** This work was sponsored by NYU Shanghai Laboratory of Urban Design and Science (LOUD); the Zaanheh Project and Center for Data Science and Artificial Intelligence at New York University (Shanghai); NYU Shanghai Major-Grants Seed Fund (Grant No. 2022CHGuan_MGSF; sponsored by the PEAK Urban programme, supported by UKRI's Global Challenge Research Fund, Grant Ref: ES/P011055/1; Fujian Urban Investment and Technology Institute's Research Fund (Grant No. 20210201 FJCT). The funders had no role in study design, data collection and analysis, decision to publish, or preparation of the manuscript.

**Competing interests:** The authors have declared that no competing interests exist.

Wuhan, China with those infected reportedly experiencing pneumonia-like symptoms with an unknown cause. Since then, the world has experienced an unprecedented shakeup as the virus began to spread around the world, rapidly infecting over 200 countries and territories with casualties numbering over 4 million as of July 2021. The viral outbreak has overwhelmed an increasingly global healthcare system, leading to shortages of personal protective equipment and crucial life-saving medical systems, while throttling local and global economies alike. As a result of its increasingly digital nature, communication via micro-blogging social networks have imbued the study of human behavior, including individual sentiment, group topics, and even identity-politics with a big-data driven science. Twitter, with over 199 million monetizable daily active users (mDAU) generating over 500 million tweets per day as of Q1 2021, has historically served as a reliable source of social expression, largely as tweets tend to contain the following useful properties: textual data (topics), temporal data (time-series component), and spatial data (geo-tagging and profiles) [4]. As of June 2019, Twitter has made geo-tagging an opt-in feature, meaning users must actively request Twitter include their locations in Tweets (only 1–2% of tweets are now geo-tagged as a result).

As is the case with any systematic shock of these proportions, individual and group reactions have been varied, with some resorting to denial, others to scapegoating, and still others to the spread and consumption of misinformation. In an effort to gauge the global pandemic's impact on social thoughts and behavior, we attempt to answer the following questions: (1) What kinds of topics are individuals and groups vocalizing in relation to the pandemic? (2) Are there any noticeable topic trends and if so how do these topics change over time and in response to major events?

The rest of the paper is organized as follows: In the literature review section, we examine the contemporary literature relating to topic modeling as it relates to twitter data and epidemiology and recent developments in Predictive Analytics and Natural Language Processing. In the research design and methods section, we describe data acquisition, data ingestion and prepossessing, and Sequential Latent Dirichlet Allocation. In the results section, we delve further into topic distribution and interpreting topic representations, before finally discussing SeqLDA limitations on topic structure and explaining unpopular topics and over-generalization. The conclusion section states how we contributed to an understanding of both the topics surrounding the COVID-19 pandemic and their evolution over time.

## Literature review

### Topic modeling

In the context of extracting topics from primarily text-based data, Topic modeling (TM) has allowed for the generation of categorical relationships among a corpus of texts, whose origins can be traced to the development of latent semantic analysis (LSA) in the late 1980's [5]. LSA itself however is really only an application of Singular Value Decomposition, attempting to identify a subspace of a Document Term Frequency Space in order to capture the majority of variance in the corpus. Latent Dirichlet Allocation (LDA), discovered by a team of University of California, Berkley researchers in 2003, unlike its discriminative counterparts, is a generative model [6]. In LDA, documents are represented by random mixtures of words over latent (i.e. emerging during the modeling process) topics. Therefore, LDA is able to identify the probability of a given document being in a given topic through a "bag-of-words" interpretation of its contents [6]. Since its emergence in 2003, LDA has played a benchmark role as a model for TM, and has since seen various domain-specific improvements and adjustments. In 2006, Blei and Lafferty (2006) [7] introduced a temporal component to topic modeling, referred to as Sequential Latent Dirichlet Allocation (SeqLDA), which focuses on modeling how topics in

a corpus of documents evolve over time by instead utilizing a state space model to chain topic and word distributions over time. Further enhancements to LDA include Online LDA, developed by Hoffman et al. (2010) [8], which implemented an online variational Bayes algorithm to appropriately scale the training process to big data objectives. In 2012, Zhang and Sun (2012) [9] added a parallel probabilistic generative model to LDA, which included correlations between users to generate topics and saw modest improvements to accuracy. At the same time, Huang et al. (2012) [10] attempted to simplify the feature space by first implementing a single-pass clustering algorithm, before utilizing traditional LDA on the new vector space. The year after, Yan and Zhao (2013) [11] furthered the micro-blog topic domain by utilizing a greater feature space relating to micro-blog posts, focusing however on a simplified LSA model. Table 1 depicts a summary table containing the contemporary variations of LDA.

## COVID-19 pandemic and social media: Twitter in predictive analytics

The outbreak of the Novel Coronavirus Diseases (COVID-19) has spread across the globe since late 2019. It caused significant impacts on people's daily life and taken hundreds of thousands of lives away [12]. The lockdown and vaccination policies also affected billions of people and the impacts on global economy, transit, and public health are profound [13]. According to WHO, as of February 2022, there are over 430 million confirmed cases, 5.9 million confirmed deaths, and more than 10 billion vaccine doses administered (WHO, 2022). Nonetheless, this is not the first-time mankind is facing a pandemic. The study of Epidemiology has aided in the fight against diseases for over 100 years [14], helping to evaluate, isolate, and stem their presence in human populations [15]. In recent decades, the increasing frequency of viral outbreaks, disproportionately impacting emerging countries, has led to calls for a broader response to public health crises, incorporating disciplines including Government, Communications, Social Sciences, Environmental Policy, Urban Studies, and Data Science—the key areas involved in a comprehensive response to a global pandemic [15–19].

Recently, the advent of social media has further enabled the behavioral study of how individuals and groups think about and react to viral outbreaks and their responses through real-time data on population sentiment [20–23]. Prior to the growth in popularity of social media outlets such as Twitter, search engine queries provided a possible data source to predict Influenza-Like-Illness (ILI) rates [24]. However, due to insufficient contextual information in raw search queries, the relation between search query activity and ILI rates remained difficult to establish and proposed models performed poorly on unseen data [23]. The contextual data that search queries lacked was augmented with access to social media data, such as tweets,

**Table 1. Recent literature on topic modeling: LDA advancements and variants.**

| Author | Year | Variant | Description |
|---|---|---|---|
| Blei et al. | 2003 [6] | LDA | Original Generative Model |
| Blei and Lafferty | 2006 [7] | SeqLDA | First Dynamic Topic Model |
| Hoffman et al. | 2010 [8] | OnlineLDA | First Scalable LDA Algorithm |
| Zhang and Sun | 2012 [9] | MB-LDA | Feature Space Expanded to User Network |
| Huang et al. | 2012 [10] | - | Clustering Feature Space prior to LDA |
| Yan and Zhao | 2013 [11] | MB-LSA | Expanded Micro-Blog Feature-Set + LSA |
| Wang et al. | 2016 [62] | SH-LDA | Updated Temporal & Hashtag-Graph-Based Topic Model |
| Xu et al. | 2016 [63] | TUS-LDA | Joint Temporal and Emotional Probability Space LDA |
| Yao and Wang | 2020 [57] | - | 3-Step Geo-Topic Generation and Tracking LDA |
| Du et al. | 2020 [60] | MF-LDA | Analyzed the life-cycle of "hot-topics" with Dynamic LDA |
| Tan and Guan | 2021 [61] | - | Recognized time and space frequency patterns |

providing for more structured content (e.g. hashtags, mentions, user-to-user interactions etc.) with greater descriptive attributes. Literature concerned with predicting localized ILI rates focusing on symptom-related statements emerged with Lampos and Cristianini (2010) [25] obtaining a 95% correlation with data from the UK's Health Protection Agency. For example, Signorini et al. (2011) [23] developed a model for predicting disease activity in real time on a national level. Broniatowski et al. (2013) [26] building a pipeline to distinguish disease-relevant tweets from online chatter and improving predictive accuracy on unseen data. Further refinement on detecting the subjects of tweets (i.e. user vs. family members) was undergone by Yom-Tov et al. (2015) [27], who built a model for predicting the Secondary Attack Rate (SAR) and Serial Interval (SI) of influenza outbreaks in the UK.

The primary application of social media data in the study of pandemic is in predictive analytics: to be able to predict, to varying degrees of spatial resolution, a metric concerning a disease [23, 28–30]. As ethical concerns over data privacy continue to advance [31–34] Twitter and other social media platforms have incrementally restricted access to certain metadata attributes as well as made certain more invasive data practices, such as geo-location opt-in [35, 36]. The result is such that many historical works dependents on data with specific meta attributes, such as geo-coordinates, now consist of only small subsets of the population (i.e. only 1–2% of the total population) and these sub-populations tend to be very biased towards a younger demographic and commercial uses of Twitter [37–39].

Although various works exist on the topic of reverse engineering twitter data, including content-based approaches [39–41] meta data approaches [42], and hybrid approaches [38, 43–45], these methods are outside the scope and focus of this work. We focus our analysis on the entire-population of Twitter users. One of the earliest TM techniques applied to epidemiology using Twitter data was Paul and Dredze (2011a) [46] Ailment Topic Aspect Model (ATAM), which identified isolated ailments such as influenza, infections, and even obesity, from a collection of 1.6 million tweets, extending LDA with a secondary latent ailment variable, to better bucket diseases. The model was improved upon with the development of ATAM+ with an added predictive component, built on a more medically-specific corpus of hand-picked articles relating to specific diseases [47]. With the advent of modern topic modeling and clustering techniques, works focused on more comprehensive examinations of user behavior began to surface. Despite the lack of geo-tagging capabilities in modern applications of Twitter data, there remains literature related to topic extractions outside of a geo-specific context. As early as 2011, Signorini et al. (2011) [23] utilized Twitter data to track public concern of the spread of the 2009 Influenza A H1N1 Pandemic, identifying strong influxes of topics revolved around hand-washing and mask safety. Roy et al. (2019) [48] focused instead on understanding the overall proximate blame tendency of online users on Twitter, with relation to the Ebola Outbreak, utilizing a sequential LDA model to follow the evolution of localized blame in a retroactive study of the outbreak. Ahmed et al. (2019) [20] implemented a similar technique in a study of topics surrounding the H1N1 pandemic, identifying a subset of misguided Twitter users that believed pork could host and/or transmit the virus. The 2015 Zika outbreak led to an examination of millions of tweets geolocated in North and South America by Pruss et al. (2019) [49], who discovered increases in public attention to Vaccinations, Viral Testing, Symptomatic Topics, and increases in polarizing political topics.

The ongoing pandemic of COVID-19, especially the Delta variant and Omicron variant, prompted urgent needs for more in-depth studies of COVID-19 and Twitter in predictive analytics. For example, in a study by Rajput et al. (2020) [50], Tweets posted by both social media and WHO were investigated. They found more positive responses to COVID-19 than negative emotions. Other studies explored the associations between COVID-19 and the human mobility restrictions, lockdown, and social distancing and on limiting the spread of the virus [51–

53]. More recently, Twitter data has been leveraged to help understand the utility of public sentiment and concerned topics in public health. Early work in 2020 by Boon-Itt and Skunkan (2020) [54] implemented topic models against a small subset of tweets spanning the first months of the pandemic, identifying multiple stages of public awareness as the virus spread, as well as different vocabularies associated with positive and negative sentiments on the outbreak. Jang et al. (2021) [55] examining the relationship between public health promotions and interventions and public perceptions in Canada, leveraging an aspect-based sentiment analysis (ABSA) model to generate topics and sentiment. Utilizing a novel LDA approach, Ahmed et al. (2021) [56] captured user-specific sentiment and sentimental clusters over time and ranked users according to their activity in trending topics, in an effort to understand user behaviors amidst varying pandemic topics.

## Contributions of this work

The focus of this work is to identify and isolate topics relating to the perception of COVID-19 as they evolve over time, focusing specifically on the behavioral epidemiological responses associated with understanding the nuances of public perception, behavior, and rhetoric on the development of COVID-19. Additionally, as the COVID-19 pandemic represents the greatest viral outbreak since the Spanish Flu, the volume of data associated with it correspondingly makes this a big-data problem, which requires a big-data lens, big-data infrastructure, and which has the advantages of accuracy that accompany large datasets. This work hopes to attain a two-fold contribution to the contemporary analysis of the most impactful virologic outbreak since the Spanish Flu, COVID-19, and to the topic modeling domain at large by focusing on the following factors: (1) A Big Data Approach to Sequential Latent Dirichlet Allocation (100 million+ Tweets); (2) A reproducible work, with a focus on an end-to-end custom reusable data pipeline; (3) An understanding of the evolution of topics surrounding the COVID-19 pandemic.

Most works concerning either theoretical topic modeling or its domain-specific application are limited by resources, the topic scope and size, and the accessibility of historical data (e.g. tweets). As a result of the size, scope, and length of the COVID-19 pandemic, the breadth of topics concerning the outbreak and consequently the number of tweets generated on the topic is exponentially greater to that of any other outbreak in recent history. Previous works reviewed on the topic of the COVID-19 pandemic have been successful in identifying relevant topics for public health purposes, but have limited themselves to smaller datasets over broader periods of time. This work has focused on delivering an end-to-end scalable Topic Modeling Pipeline, with a publicly accessible GitHub repository (https://github.com/akbog/urban_data) outlining methods and technologies used, successfully achieving a scale of millions of analyzing millions of tweets per day. Additionally, our work leverages dynamic versions of LDA to measure topic drift on a topic and vernacular level, helping to identify changes in trending topics at scale and dynamically over time.

## Research design and methods

### Data acquisition

Twitter has historically remained a relatively open platform for data analytics, making it a popular source of public opinion and thought amongst academics. Yet there are still certain hurdles faced when acquiring twitter data samples, namely streaming limitations and more recently geo-location restrictions. See Table 2 for a description of Twitter's Developer API Services and Limitations.

**Table 2. Twitter API services & limitations.**

| Twitter API Access | | |
|---|---|---|
| **Service** | **Tweets/Month** | **Description** |
| 30-Days Sandbox | 25k | Tweets only available from within the last 30-days |
| Full Archive | 5k | Tweets from the full twitter archive (since 2008) |
| Standard Stream | Rate-Limited | Stream Live Tweets from the last 14-days (Excessive requests can generate rate-limits) |

Because of the data-size limitations posed on the 30-Days Sandbox and Full Archive datasets, as well as the on-going problem we are studying, we decided to steam as many tweets as possible (24/7 streaming through Twitter's API Endpoint) starting from March 31$^{st}$, 2020. The data collection method complied with the Twitter terms of service (The Twitter rules and policies can be found here: https://twitter.com/en/tos). Twitter's Streaming API allows for several request parameters that can filter for language, location, and key-words. Specified conditions act as "OR" statements, not "AND", meaning specifying a location and key-words produces geo-tagged tweets from a specific location or matching a given key-word. As we are only concerned specifically with tweets pertaining to COVID-19 we use the following key-words to filter COVID-19 related tweets during the streaming process:

["coronavirus", "corona virus", "covid19", "covid", "covid-19", "wuhan", "sars-cov-2", "pandemic", "epidemic", "outbreak", "virus", "infect"]

Additionally, after a connection is broken or stalled, Twitter's API may return duplicate tweets in each request. We therefore build dictionaries to test whether a given tweet_id (which is a unique identifier) has already been streamed prior to outputting into files. We output files into compressed gzipped bundles of approximately 5,000 tweets (gzip is a file format and a software application used for file compression and decompression).

## Data ingestion and preprocessing

**Data structure and apache spark.** By default, tweets are outputted into the familiar, dictionary-like, JSON format. The main component to a tweet object is the tweet itself and the user object, with certain tweets containing extra retweet, quote, or reply objects depending on the type of tweet. A theoretical entity relationship database structure is presented in Appendix A. The dataset was scraped over a 14-day period starting March 31st and is composed of over 46 million tweets, averaging over 3 million tweets per day, before preprocessing. We initially set out to ingest and preprocess the dataset by streaming it into memory, procedurally processing it, and outputting it to disk storage. We leveraged and extended portions of Natural Language Toolkit (NLTK)'s python natural language pipeline (https://github.com/nltk/nltk) as well as those found in Gensim's (https://github.com/RaRe-Technologies/gensim) library (e.g. TextNormalizers, NGram Generators, Vectorizers etc.) to produce preliminary results. However, the time complexity of these python-based libraries was a serious barrier given the size of our dataset, python's interpreter and memory, and resource constraints. To this end, we took advantage of the popular open source distributed big data framework, Apache Spark (https://github.com/apache/spark), which provided us with the necessary speed and scale (i.e. distributed, in memory, parallel computations), a convenient high-level python api (i.e. PySpark), and a powerful optimized natural language processing library (i.e. John Snow Lab's SparkNLP https://github.com/JohnSnowLabs/spark-nlp). This allowed us to build an end-to-end data pipeline as can be seen in Fig 1.

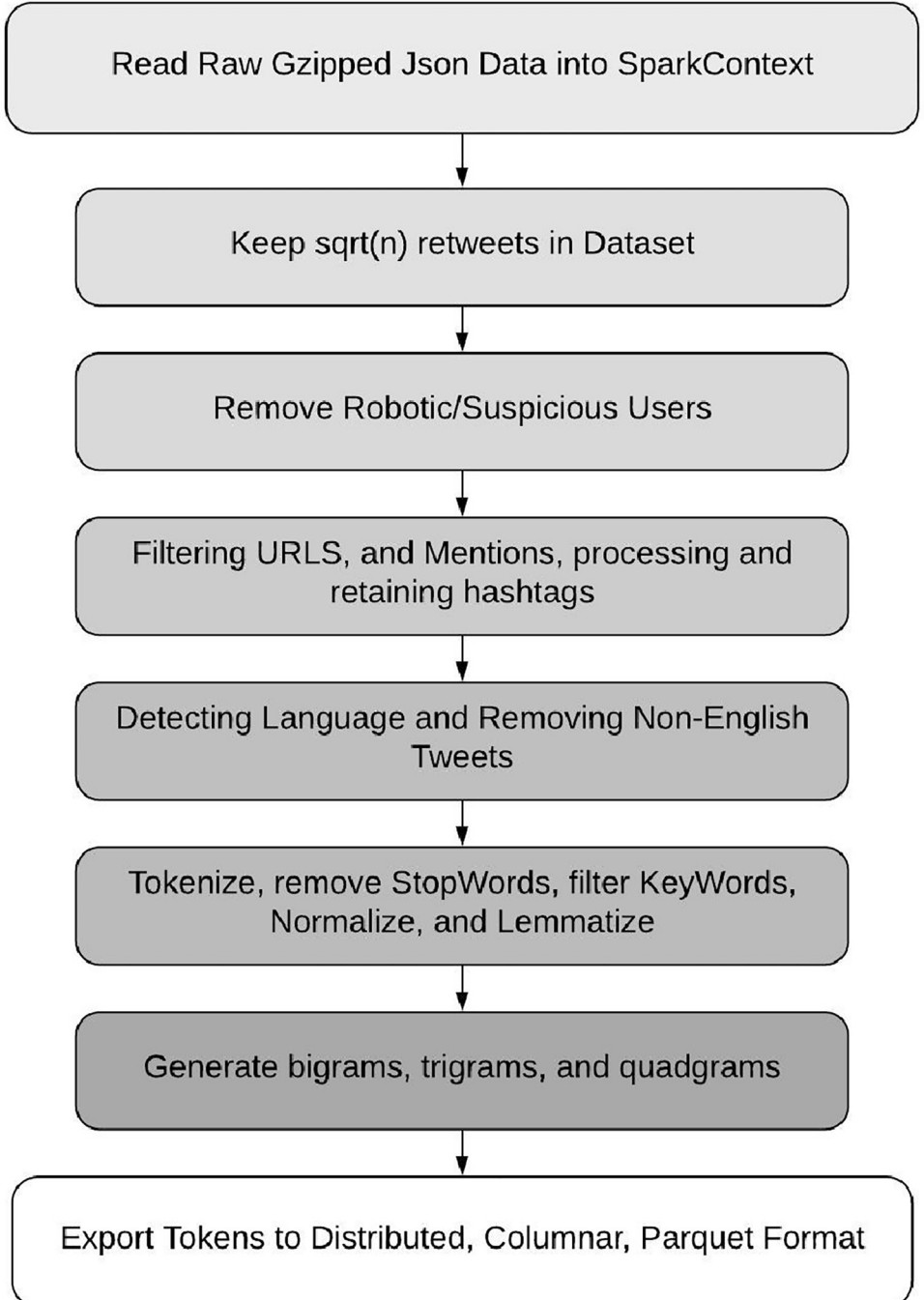

**Fig 1. Dataset preprocessing stages.**

**Re-scaling retweets.** In the case of LDA, we must begin by transforming our semi-structured dataset of random words (i.e. Tweets) into a machine-learning readable format. We begin by understanding that, as it stands, over 50% of our dataset of tweets is in the form of structured retweets. On Twitter, retweets represent an opportunity for users to share each other's tweets, increasing outreach and discussion, meaning they can be useful as a proxy for user expression, despite their indirect nature. As a result, we refrain from purging retweets from

our dataset and instead focus on re-scaling the population of retweets as seen in Lozano et al. (2017) [38], by reducing their count to $\sqrt{N}$ of the number of retweets, $N$ present our dataset.

**Removing robotic & suspicious accounts.** As proposed in Yao and Wang (2020) [57], we consider the possibility that certain users may exhibit features outside the realm of normal twitter use (i.e. excessive tweets, retweets, followers etc.). Hence, we deploy a similar method to remove users—and therefore tweets—whose Z-score is outside three standard deviations from the norm, calculated as follows:

$$Z_{(\text{tweets,followers,friends})} = \frac{\log 10 V(\text{tweets, followers, friends}) - \mu(\text{tweets, followers, friends})}{\sigma(\text{tweets, followers, friends})} \quad (1)$$

We take the log of each value (i.e. tweets, followers, friends) to account for their extremely right skewed distribution.

**Cleaning text & removing non-English tweets.** It is important to remove trivial and non-semantic texts such as mentions and uniform resource locators (urls), given their tendencies to 'explode' the vocabulary size and their lack of semantic intents. As hashtags often contain popular or trending words or phrases, we split the hashtags in place such that hashtags of the format "#StayHomeCovid19" were transformed into "Stay Home Covid 19". The grammatical structure of the texts was retained to guarantee the best language detection in latter stages. We utilized the popular langdetect library (https://github.com/Mimino666/langdetect), a port of the java language detection algorithm developed by Nakatani Shuyo (2010) [58], supporting 55 languages, to filter out all non-English texts. The Naive Bayesian nature of the algorithm means it is both fast and scalable, with a precision of 99.8%. We found that, on our dataset of short texts (i.e. tweets), the langdetect library outperformed the native SparkNLP pretrained LanguageDetectorDL annotator and attributed this to the diversity of langdetect's training and testing dataset, whereas the SparkNLP annotator was exclusively trained on longer texts sourced from Wikipedia. In order to integrate langdetect into our PySpark workflow, we implemented user-defined functions in Spark, and passed the langdetect algorithm to our distributed dataset. Fig 2 depicts the resulting tweet distribution after removing retweets, users with abnormal behaviors (e.g. bots), and non-English texts.

**Standard NLP cleaning.** In standard NLP fashion, we tokenize our texts, splitting on white space, and use a list of popularized StopWords, compiled from multiple sources (i.e. SpaCy, SparkNLP, NLTK), combined with a custom set of words that have shown to be common across topics to clean our texts. For example, given that a pre-requisite of membership in the dataset is a key-word match with our key-word dictionary and given that COVID-19 goes by many different names (e.g. Coronavirus, Sars-Cov-2 etc.), we clean our dataset of all

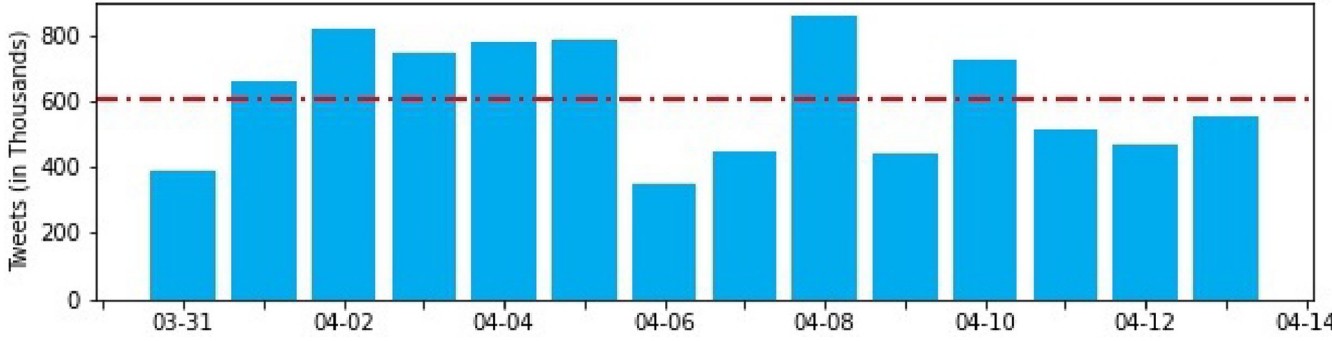

**Fig 2. Distribution of preprocessed Tweets.**

common references to the virus. During model training, certain terms such as "time", "need", or "day" proved to have dilutive properties, appearing in many topics and causing topic convergence. We carefully consider these terms at this stage, and in later stages during vectorization and vocabulary creation. Additionally, we normalize (i.e. standardize case and remove punctuation) and lemmatize (i.e. revert to common base word) each tweet utilizing SparkNLP's Normalizer, Lemmatizer, and Stemmer annotator classes (https://nlp. johnsnowlabs.com/docs/en/annotators) to produce a standardized vocabulary.

**Generating ngrams & storage.** Ngrams are sequences of words that tend to appear next to each other frequently throughout the dataset. It can be useful to identify these pairs, triplets, or quadruplets (i.e. bigrams, trigrams, or quadgrams) as they can be more intuitive, topic-wise, than their individual components, and can help distinguish words that co-occur in topics, from words that make up the same concept or phrase. In order to generate our ngrams, we utilize SparkNLP's NgramGenerator to take in our ordered and preprocessed tokens, outputting all pairs, triplets, and quadruplets. We then build CountVectorizer models, which consider a minimum threshold of ngram appearances in tweets as a percentage of the dataset for inclusion in our vocabulary. The ngrams that meet these thresholds are substituted back into each tweet, in order to prevent both the presence of ngrams and the words that compose them from being present at the same time in each tweet.

After all stages of preprocessing are completed, we export a tokenized version of our dataset, stripped of meta-attributes, into a native Apache Spark schema-oriented binary-based columnar file system known as Apache Parquet (https://parquet.apache.org/documentation/latest/), which offers significant scalability, as well as efficient read and write times on our final tokenized dataset.

**Vectorization.** In order to work with LDA models, we first vector encode our tokenized tweets into a one-hot encoded term-frequency matrix with m rows (i.e. # of tweets in our corpus) and n columns (i.e. size of our vocabulary). We trained a Spark CountVectorizer Model, which encodes each tokenized tweet as a sparse vector, where the presence of a word is encoded in binary (i.e. 1 for present, 0 otherwise). Whereas we tested Term Frequency—Inverse Document Frequency (TF-IDF), term weighting strategy developed in the early 1970s and still used in the majority of NLP applications today [59], and normal (absolute) vector encodings of the dataset, the decision was made to utilize one-hot vector encoding as a result of two factors; our dataset is made up of short (typically <140 character) tweets which do not typically repeat terms and as LDA is a word generating model, TF-IDF score representations are not readily interpretable. Additionally, the CountVectorizer Model excludes words appearing above and below a specified threshold of presence. If a word is present in greater than 50% of the dataset or in less than 0.08% of documents, we exclude them from our vocabulary. This is done to catch any stop words we may have missed while preprocessing, to ensure that no single word becomes dominant across too large a sample of topics, and to prevent our vocabulary from becoming excessively large.

## Sequential latent dirichlet allocation

Sequential LDA was first discovered by the original co-creators of LDA, Blei and Lafferty (2006) [7]. Whereas many incremental changes have been made to LDA since then—as described in earlier—the dynamic component of the original 2006 SeqLDA work is sufficient for our purposes. In normal static LDA, documents are represented as random mixtures of latent topics, where each topic is characterized by a distribution of word probabilities. The plate diagram shown in Fig 3 displays the process for generating M documents (i.e. tweets), each with N words. $\beta$ depicts a probability matrix representing the probability of a word being

in each topic, whereas $\alpha$ is a vector representing the probabilities of a given topic being present in a given tweet. Both are hyperparameters that effectively establish the Dirichlet distributions that govern the model. In essence, classic LDA is a method of attempting to understand what words make up which topics and how these topics make up a document through words. Sequential LDA provides static LDA with a dynamic component by utilizing a state space model, as depicted in Fig 4, which replaces the Dirichlet distributions with log-normal distributions with mean $\alpha$, chaining the Gaussian distributions over K slices and effectively tying together a sequence of topic-models. To implement the logic outlined in Blei and Lafferty (2006) [7], we first train a collection of normal LDA models on a subset of our data (March 31st—April 2nd, 2020) to establish the hyperparameters of our model. As Gensim already utilizes KL-Divergence to estimate $\alpha$ and $\beta$ Dirichlet priors, we only test for the optimal number of topics. Whereas Gensim's default scoring function is perplexity, we choose instead to use a measure of topic coherences which operates by maximizing the following function:

$$UMass_{(wi,wj)} = \log \frac{D(w_i, w_j)}{D(w_i)} \tag{2}$$

UMass scores higher when words appear together more frequently than they do by themselves, operating under the assumption that topics that are "coherent" will feature words that appear together more often.

The resultant models and their corresponding coherence scores can be seen in Fig 5. There was a clear benefit from increasing topic size until the number of topics reached 70, at which point there was a decided drop in coherence scores. As a result, we choose 70 Topics for our Sequential Model. As Gensim features an existing implementation of the Sequential LDA algorithm presented in Blei and Lafferty (2006) [7], we initialize our model with the pre-calculated hyperparameters, and proceed to build the model. Prior to using Gensim's implementation, we attempted two other LDA implementations: LDA Mallet, a java-based implementation with a python wrapper and guaranteed faster convergence and Sklearn. However, on the provided sample, LDA Mallet refused to converge and Sklearn's default LDA implementation proved both time-consuming and produced lower Coherence Scores than Gensim. After multiple rounds of testing, our final model took approximately 34 hours to complete, passing over the dataset 5 times each training iteration (i.e. 5 passes/time slice), updating assumptions every 1,000 tweets. A summary of our model configuration and cluster resources used can be found

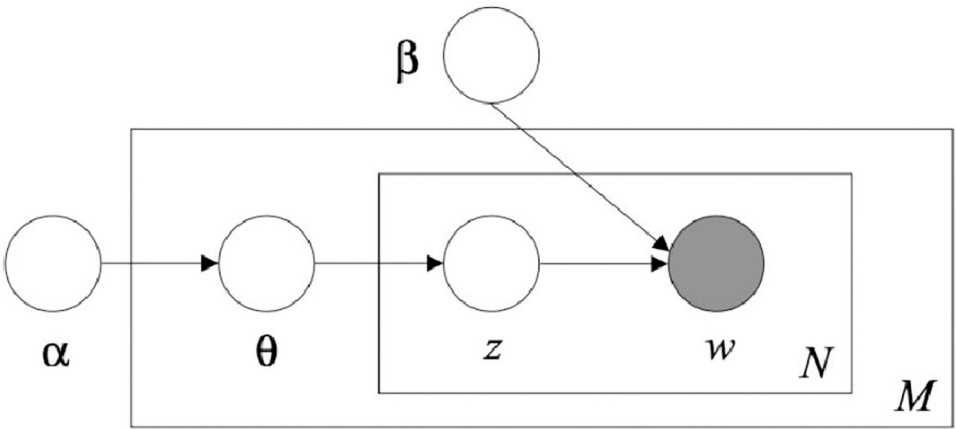

**Fig 3. Original LDA representation.**

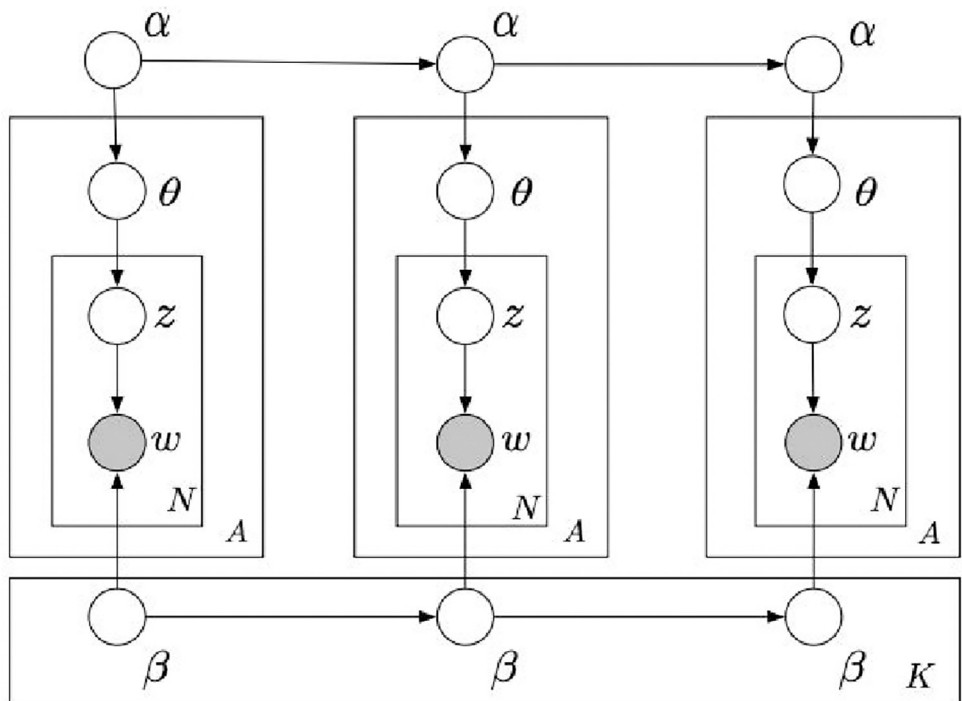

**Fig 4. Original DTM representation.**

in Table 3. We theorize that model training times could be improved through either compilation in cpython or utilizing an Apache Spark (i.e. optimized for Distributed Computing) LDA model.

## Results

### Topic distributions

As a result of the qualitative nature of working with textual data, evaluating the results of an LDA model are partly quantitative and partly qualitative in nature. This is similar to the logic presented when choosing a proper optimization function. At the end of the day, our model is designed to extract qualitative measures of the topics that individuals are discussing, and follow how these topics change over time. First, we breakdown the most popular topics present in our dataset. Whereas we optimized our model to account for 70 topics, which was derived by utilizing a grid-search strategy and optimizing for Umass, our topics do not feature equal representation in the dataset. In fact, as Fig 6 depicts, for any given day between April 3rd and April 13th, the top 12 topics over each day make up between 70 and 80% of the topics present in our dataset.

We also observe an increasing trend in the representation of the top 12 topics over the specified period. A few potential explanations could be: (1) As the Sequential Model trained over each successive time-slice, those topics making up the bulk of our dataset were well represented, note the steep increase on the first day of training; (2) There exists a strong correlation between the change in size of our day-to-day training population and the representation of the top 12 topics.

To our second point, later iterations of our training algorithm trained on smaller than average representations of the data (Due to Twitter's Rate Limits, see Data Acquisition), but

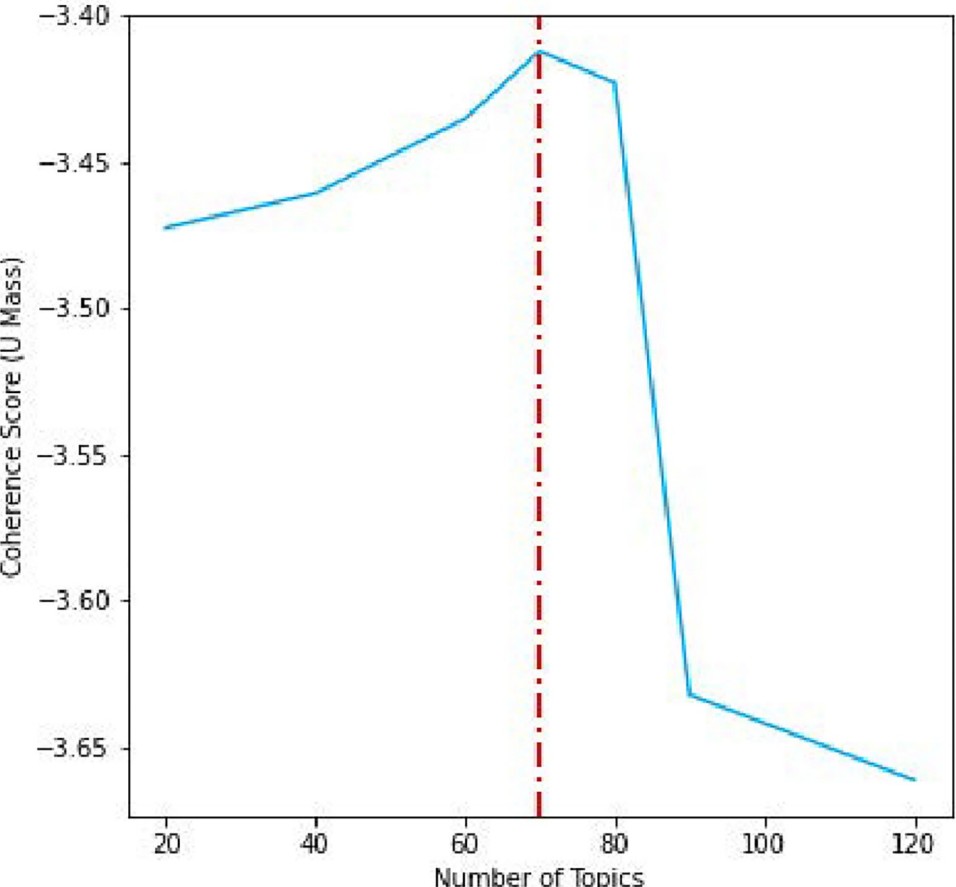

**Fig 5. Static LDA coherence scores for varied numbers of topics.**

continued to exhibit this trend. It may be that these topics are in fact "trending" on Twitter and therefore accumulating in representation over the specified period. Additionally, each of the top 12 topics exhibited coherence scores in the range from -1.12 to -3.37, suggesting that terms found in these topics co-occurred in topics to a greater extent than they appeared separately. We therefore investigated whether this was a result of terms that appeared in many topics together, or whether these terms were relatively unique to these topics.

**Table 3. Cluster & model configurations.**

| Cluster Configuration | |
|---|---|
| Nodes | 2 |
| Cores/Node | 16 |
| Memory/Node | 32GB |
| Partition | Parallel |
| **Model Configuration** | |
| Dataset Passes | 5 |
| Update Model | Every 1k tweets |
| Scoring | Umass |
| Train Time | 34 hours |

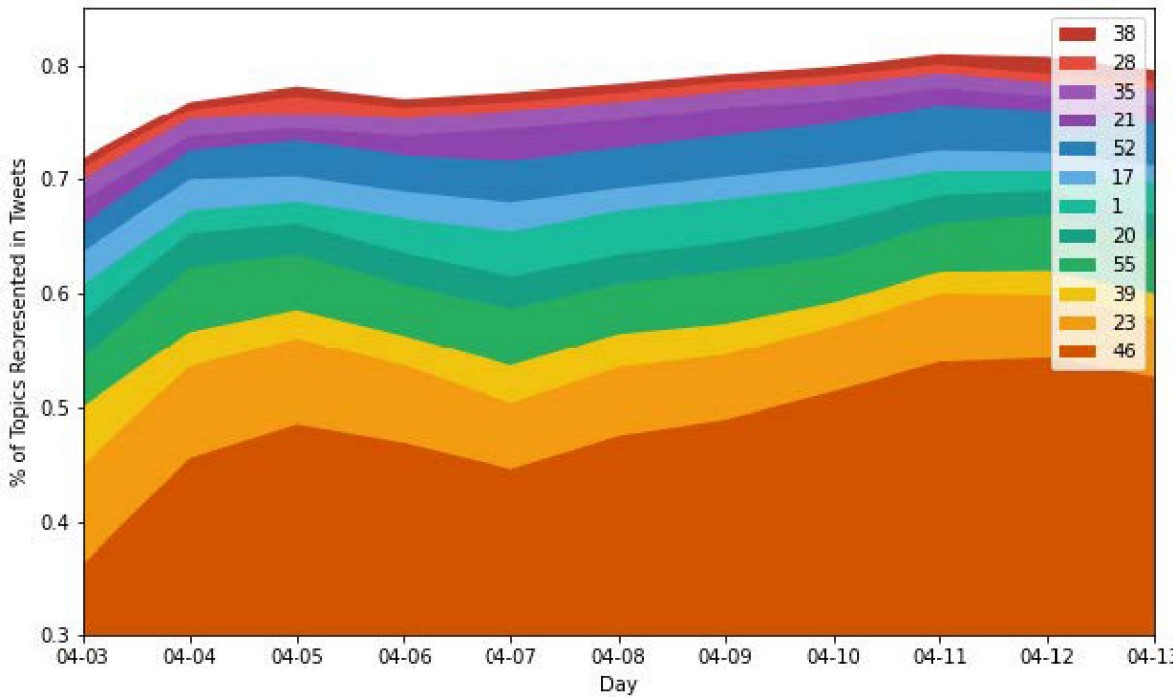

**Fig 6. Time-series representation of the changes in topic dominance over time.**

### Interpreting topic representations

We broke down each day into its respective topic representations, sampling tweets that scored high in their respective topics and providing the top ten words that best indicate a given topic. Table 4 represents the results of our labeling process as of April 13th. The 10 topics found in Table 4 are the 10 largest and combined represent over 70% of our dataset at any point in time. Almost every topic in our top ten list is unique, and all are readily interpretable, a strong sign of a successful topic model. For example, it is interesting to note that topic 20, which we labeled Social Distancing, features the word "Easter" as one of its key-words. This term was not present in topic 20 on April 3rd which indicates a greater concern about the upcoming Christian holiday and its associated social gatherings. We observe a similar phenomenon in topic 1, Personal Finances, which experiences a slight shift in importance from terms associated with small businesses, such as loans to programs related to schools and students, which is in-line with both student graduations and the government stimulus timeline. Topic 35, Medical Resources, which is similar to topic 17, Healthcare, but with a greater focus on supplies and equipment, experiences topic drift in line with ventilator deliveries, which were finally distributed in the United States in the first week of April, at the peak of the outbreak. Fig 7 summaries the topics outlined above (i.e. Easter, and Social Distancing) and demonstrates how our DTM manages to capture their relevance over time. Terms such as "student" rise in importance as Universities begin to settle into remote education, whereas terms such as "small_business" remain steady, as conversations continue to persist over the impact of COVID-19 on small businesses and the potential for stimulus. We have also come to understand that one of the largest topic pools relates to American politics and policy. Topics 55 and 39 both lead with the term Trump, with topic 55 focusing more on the upcoming election and terminology focusing on votes, democrats, and republicans, while topic 39 focusing more on terms relating to the government's response, as evidenced by the prominence of the term 'January', which is

**Table 4. Topic word representations for April 13th and custom labels.**

| Topic Interpretations | | | |
|---|---|---|---|
| Topic # | Words | Label | Topic Size |
| 46 | *time, like, need, know, world, day, life, think, going, good* | Status Updates | 52% |
| 55 | *trump, president, american, america, democrat, vote, response, china, republican, obama* | US Politics | 5% |
| 23 | *death, test, number, testing, case, vaccine, infection, rate, data, patient* | Infection & Testing | 5% |
| 52 | *case, death, new, state, new_york, total, update, county, city, reported* | Reports | 4% |
| 1 | *online, business, help, student, support, resource, free, pro- gram, new, school* | Personal Finances | 2% |
| 20 | *stay_home, stay_safe, social_distancing, safe, stay, home, lockdown, save_lives, healthy, easter* | Social Distancing | 1.5% |
| 17 | *mask, worker, nurse, ppe, hospital, patient, medical, front- line, face_mask, healthcare_worker* | Healthcare | 1.5% |
| 39 | *trump, state, january, response, election, economic, warned, american, government, warning* | American Response | 1.5% |
| 21 | *support, community, thank, help, crisis, health, response, team, excellent, time* | Positive Response | 1% |
| 35 | *supply, staff, ppe, company, equipment, worker, player, medical, employee, testing* | Medical Resources | 1% |

featured in tweets primarily discussing the delayed reaction of the United States to the virus. In fact, the Dynamic Topic Model also effectively captured the president's son-in-law Jared Kushner's increased role in the U.S. COVID-19 taskforce, which was announced on April 3rd, before commentary on his role slowly diminished over time.

## Discussion

### SeqLDA limitations on topic structure

To assess the degree to which our SeqLDA model accurately detects topics, we utilized t-distributed stochastic neighbor embedding (tSNE) clustering, a nonlinear algorithm whose purpose is to provide a low dimensional representation of high-dimensional space, while preserving the local and global distances among high-dimensional points. The intention behind the algorithm is to plot a randomly selected sample of n tweets, such that

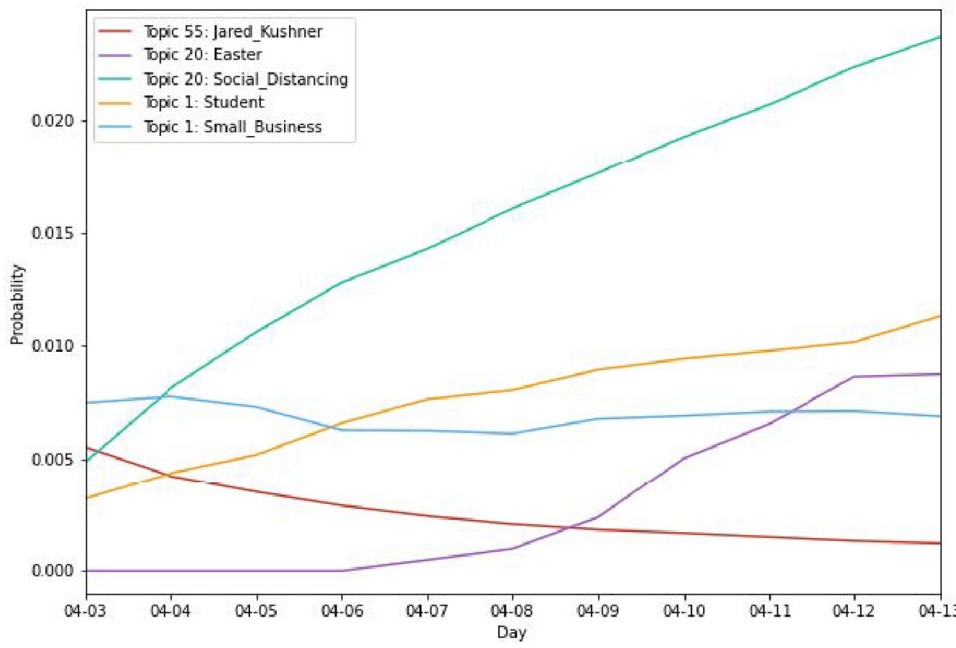

**Fig 7. Changes in topic-word probabilities over time.**

tweets with similar topic compositions will appear closer together, thereby forming clusters.

In order to distinguish between tweets, we utilize a custom hierarchical clustering coloring technique, which first assesses topic similarity based on topic-word probability vectors extracted from SeqLDA, before assigning colors on a rainbow scale to topics, based on their hierarchical relationships (see Fig 8). The reasoning is such that topics branching from the same node should be closer together on the color scale. This is done with the intention of aiding in distinguishing between tweets plotted in a lower dimensional space by coloring tweets according to the highest scoring topic.

The results of multiple rounds of training and hyper-parameter tuning are available in Fig 9, depicting the plotted tweet population sub-sample, colored by the hierarchical clustering technique. It is important to notice that, whereas our initial SeqLDA hyper-parameter optimization led us to decide on a fixed number of 70 topics to optimize for (i.e. consistency was maximized at 70 topics), the t-SNE visualization depicts a different story. As expected, documents of similar topics tended to cluster together, however, in many instances, topics separated by color did not cluster into separable groups. This evidence leads us to believe that the optimal number of topics is in fact less than 70, as many document groups tended to overlap in similar clusters. Furthermore, the dominant topic (see Section. 4.1 Topic Distributions), topic 46 tended to form many different clusters, meaning that, as discussed in Section 4.1, it does present a "catch-all" tendency, capturing documents that vary in underlying topic distribution, but that share a few key terms.

By studying our vocabulary further, we discover that a major limitation of our model is the appearance of certain common phrases that were not caught during our pre-processing stages. Whereas our topics still retained a high-degree of coherence and interpretability, the following set of words had a dilutive impact on our topics:

*["time", "need", "like", "day", "today", "update", "help"]*

For demonstrative purposes, Fig 10 is the result of a SeqLDA model trained on a single day, with similar configurations. We exclude words with dilutive properties and limit the number of topics to 30, as we found the tradeoff between overlapping topics did not warrant the small gain in Coherence score (see Fig 4), once the list of dilutive terms had been excluded. The results clearly demonstrate the impact that these groups of words, as well as the number of topics, has on the structure of our document groups. It's clear that the new visualization is able to

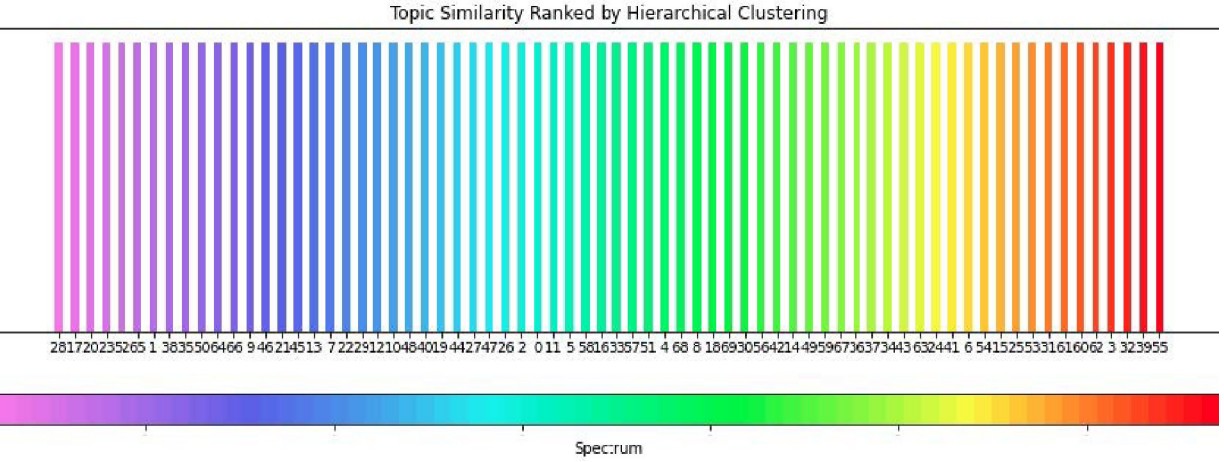

**Fig 8. Custom hierarchical coloring.**

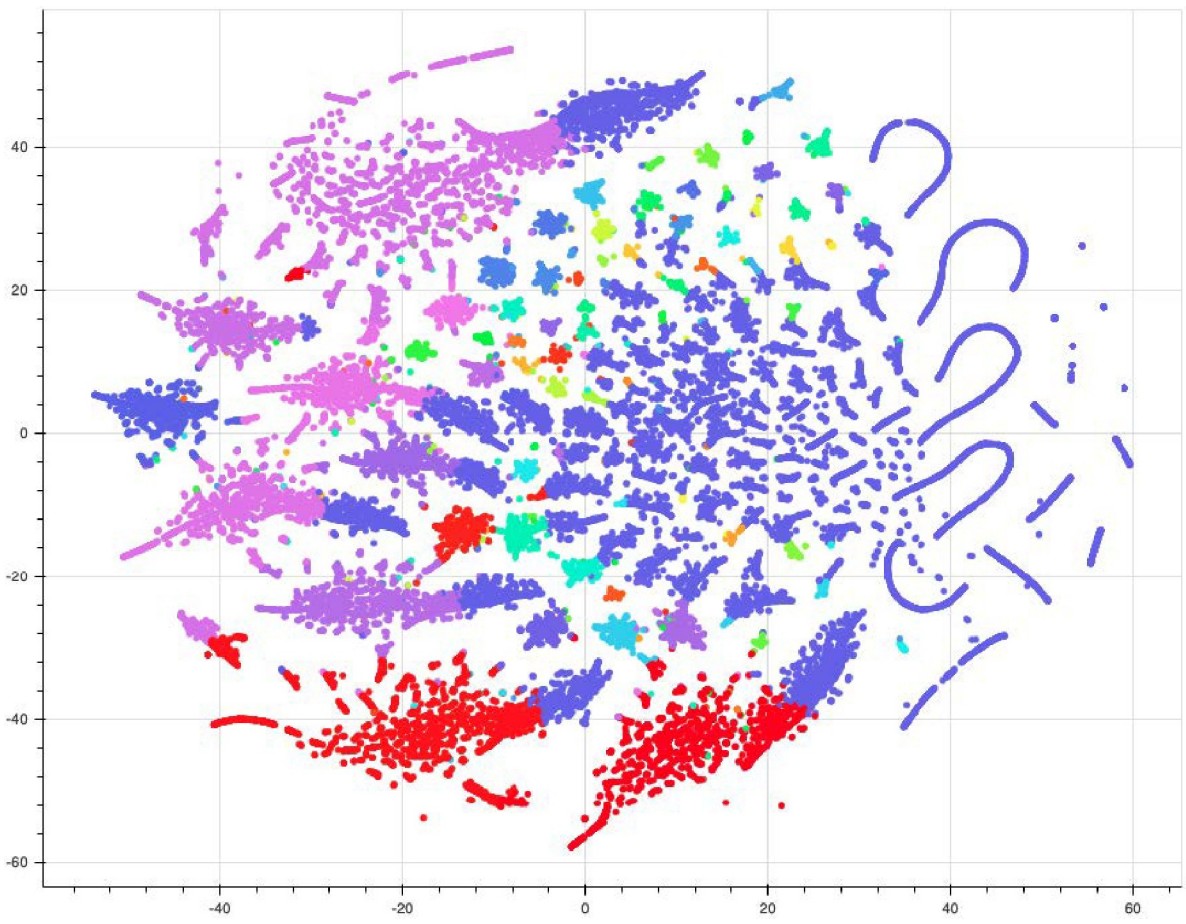

**Fig 9. t-SNE clustering visualized.**

cluster 30 topics into distinguishable non-overlapping groups without these terms, and this strengthens the original intuition of topic 46, which encompassed greater divisions of our topic space thanks to the general usage of terms such as "time", "need", or "day". Whereas we emphasize that our initial model retained highly coherent and interpretable topics, we also understand its limitations and weaknesses.

## Unpopular topics & over-generalization

While we have discussed the state and strengths of our model, specifically as it is able to effectively and intuitively capture term and topic trends over time, it is important to discuss certain weaknesses of this Sequential LDA implementation and discuss some less-popular (as a percentage of Dataset) categories. To begin with, the most-popular category, at times with presence in approximately 50% of tweets, which we have labeled as Status Updates due to the general nature of the corpus of words that represents it, is a bit too general. We note that terms such as "time", "like", and "need" tend to appear together in tweets and therefore in our topics. Our reliance on topic coherence, a standard practice in LDA modeling, may skew the proportion of tweets that belong to this category by scoring it higher as a result of the co-occurrence of these terms. This is confirmed by the degree to which these terms occur in other topics, relative to the other top topic categories.

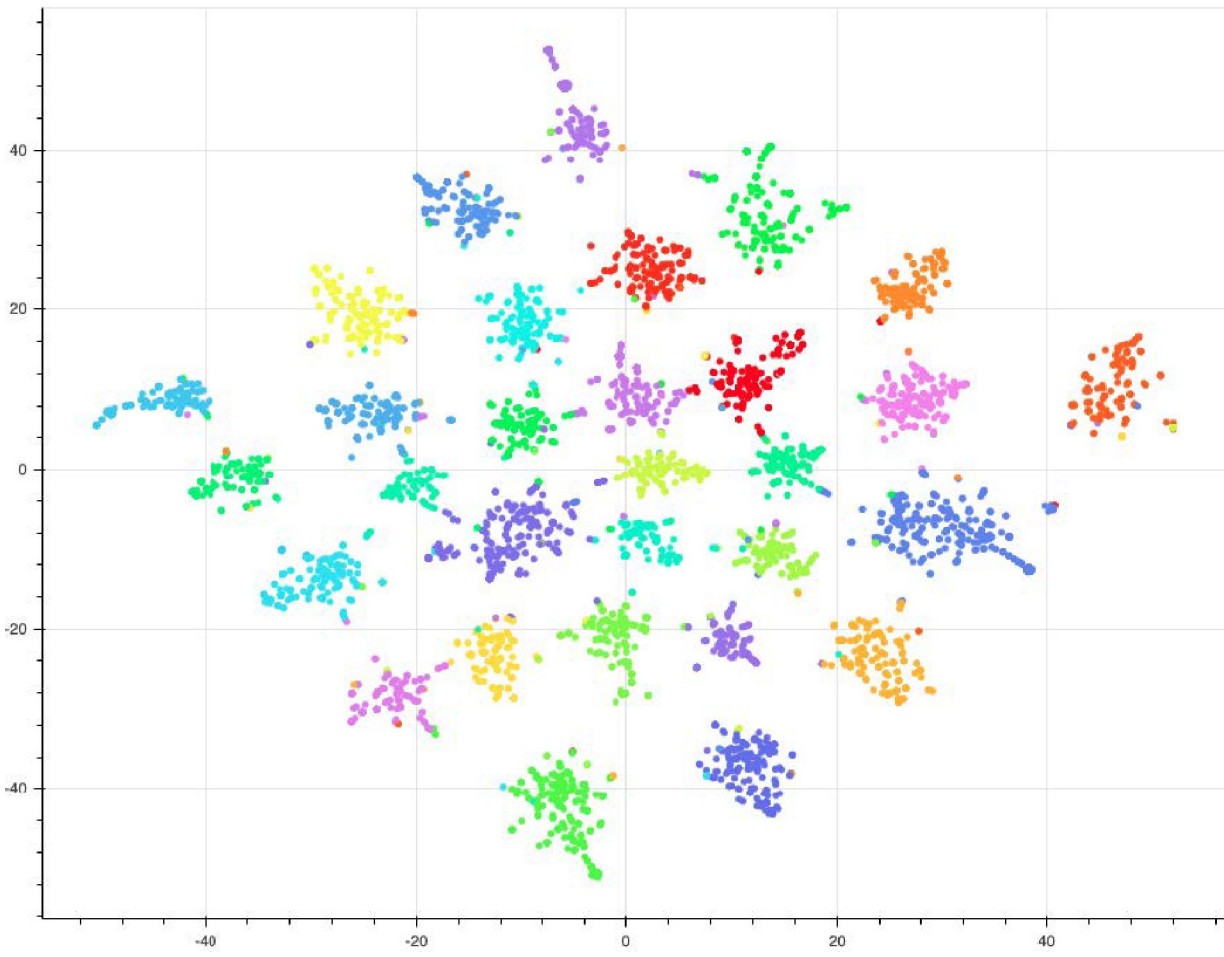

**Fig 10. t-SNE sample clustering visualized.**

A potential solution for this issue is to add a stop-word filter for terms such as "like", "going", or "think", but we must do so with a high degree of confidence that these terms do not in-fact relate to any latent topics. For instance, the term "think" might express a larger amount of self-expression as compared to other categories (hence our Status Update Label). Topic models are designed to splice documents into their root topics through their representational word probabilities. In this case, we should ask whether the relative size of the topic is proportional to its contribution to the document. If not, as may be the case here, removing certain key-terms of these topics may benefit the model.

We must also point out that, although certain topics within the model evolve well over time, other smaller topics, which cover more niche but still widely discussed subjects have more drastic evolutions over time. Because we are working in the time-span of weeks, topics are more likely to rise and fall, with their word-topic probabilities evolving accordingly. This is also a detriment of considering a fixed number of topics from the beginning, which amounts to a pseudo-zero-sum effort among topics. In doing so, we limit emerging topics to a pre-existing and fixed topic-word space, forcing existing topics to potentially change as a result. Major topics presented in this study however, do exhibit relatively consistent trends over time and tend to encapsulate domains rather than events effectively.

## Conclusion

At the beginning of this study, we set out to build a working dynamic topic model to be applied to a large and growing dataset of tweets specifically concerning the COVID-19 pandemic. We demonstrated a reproducible, robust technical solution that spanned the entire data processing pipeline, from data acquisition to data modeling, covering an online storage solution and thorough preprocessing, tokenization, and vectorization efforts in between.

Our approach differentiated itself in both scale and scope, utilizing advanced SeqLDA to study the emerging topics concerning COVID-19 at a scale that few works have been able to achieve, but that many will be able to reproduce, given the open source and architecture heavy nature of our research. By grasping the topic early, we were able to stream a sufficiently large corpus of tweets live (measuring in the 100's of millions), building a domain-specific corpus to be used in both current and future works. In this way, we contributed to the cross-sectional field of Urban Research and Big Data.

Through our SeqLDA model, we contributed to an understanding of both the topics surrounding the COVID-19 pandemic and their evolution over time. Specifically, we identified 12 of the most popular topics present in our dataset over the period spanning April 3rd to April 13th 2020 and discussed their growth and changes over time. These topics were both robust, in that they covered specific domains, not simply events, and dynamic, in that they were able to change over time in response to rising trends in our dataset. They spanned politics, healthcare, community, and the economy, and experienced macro-level growth over time, while also exhibiting micro-level changes in topic composition.

Whereas we are optimistic towards the future, we also understand that this is an unprecedented time that will have lasting impacts on individuals and society at large, impacting not only the economy or geo-politics, but human behavior and psychology. Therefore, in more ways than one, this research is just beginning to scratch the surface of what will be a concerted research effort into studying the history and repercussions of COVID-19.

## Supporting information

**S1 Appendix. Entity-relationship database structure diagram.**
(PDF)

**S2 Appendix.**
(DOCX)

## Author Contributions

**Conceptualization:** Alexander Bogdanowicz, ChengHe Guan.

**Data curation:** Alexander Bogdanowicz.

**Funding acquisition:** ChengHe Guan.

**Investigation:** ChengHe Guan.

**Methodology:** Alexander Bogdanowicz.

**Project administration:** ChengHe Guan.

**Resources:** ChengHe Guan.

**Software:** Alexander Bogdanowicz.

**Supervision:** ChengHe Guan.

**Validation:** Alexander Bogdanowicz.

**Writing – original draft:** Alexander Bogdanowicz, ChengHe Guan.

**Writing – review & editing:** Alexander Bogdanowicz, ChengHe Guan.

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
