## [Decision Letter · Decision Letter 0]

26 Oct 2021

PONE-D-21-27183Public sentiment during the COVID-19 pandemic: Dynamic topic modeling of Twitter data in the United StatesPLOS ONE

Dear Dr. Guan,

Thank you for submitting your manuscript to PLOS ONE. After careful consideration, we feel that it has merit but does not fully meet PLOS ONE’s publication criteria as it currently stands. Therefore, we invite you to submit a revised version of the manuscript that addresses the points raised during the review process.

We look forward to receiving your revised manuscript.

Kind regards,

Kazutoshi Sasahara

Academic Editor

PLOS ONE

Additional Editor Comments (if provided):

Both reviewers think the manuscript has a merit for publication; however, they also think it needs more wok. Please read the comments and revise accordingly.

Journal Requirements:

2. In your Methods section, please ensure that you have included a statement specifying whether the data collection method complied with the Twitter terms of service.

“This work was sponsored by the Zaanheh Project and Center for Data Science and Artificial Intelligence at New York University (Shanghai); Fujian Urban Investment and Technology Institute’s Research Fund (Grant No. 20210201 FJCT).

“This work was sponsored by the Zaanheh Project and Center for Data Science and Artificial Intelligence at New York University (Shanghai) The funders had no role in study design, data collection and analysis, decision to publish, or preparation of the manuscript.”

Reviewers' comments:

Reviewer's Responses to Questions

**Comments to the Author**

1. Is the manuscript technically sound, and do the data support the conclusions?

Reviewer #1: No

Reviewer #2: Yes

2. Has the statistical analysis been performed appropriately and rigorously? 

Reviewer #1: I Don't Know

Reviewer #2: N/A

3. Have the authors made all data underlying the findings in their manuscript fully available?

Reviewer #1: No

Reviewer #2: Yes

4. Is the manuscript presented in an intelligible fashion and written in standard English?

Reviewer #1: No

Reviewer #2: Yes

5. Review Comments to the Author

Reviewer #1: The articles have several issues which I want to present here.

One main issue is that whether it is title, abstract, or research questions everywhere the authors have talked about public sentiments but no sentiment analysis has been performed in this research paper. This research paper is only presenting the results from topic modelling. This is very confusing. I don't understand why the authors is written about public sentiment when they are not performing sentiment analysis.

1. Introduction

Organization of paper can be more specific (rewrite it in a better way and as a separate paragraph)

2. Literature Review

Literature Review is vague. It should be focused on the related works that include topic modeling, COVID-19, Social media (Twitter).

This line - As of June 2019, Twitter has made geo-tagging an opt-in feature (only 1-2% of tweets are now geo-tagged) - is not clear.

Section 2.4 Contribution of this work

This have a spelling mistake in Spanish Flu which is written as Spanish Flue. Also, novelty of the work can be explained in better way (improvement needed)

3. Research Design and Methods

Section 3.2 Data Ingestion and Preprocessing

It is not clear whether data in this Line - The dataset was scraped over a 14-day period starting March 31st and is composed of over 46 million tweets, averaging over 3 million tweets per day. - is before preprocessing or after preprocessing.

4. Results

Section 4.1 Topic Distributions

Second paragraph is not incomprehensible.

Section 4.2 Interpreting Topic Representations

Table 4 shows ten topics which I guess are the biggest ten topics but how can we know this? Topic weightage, or topic size usually shows this but this research paper is not showing any of this. I suggest the authors to show in table 4 topic size also.

How (on what basis/criteria) the 5 topics in Figure 7 is selected?

5. Discussion

Section 5.1 SeqLDA Limitations on Topic Structure

Authors selected 30 topics but they did not provide any explanation about how they came up with this number?

6. Conclusion is not well written.

Lastly, authors have written that the data of this research paper is available at NYU Shanghai Library but they provided nor access number or link to access this data. They should make all the data available so that reviewers can access it and verify this research's authenticity.

Reviewer #2: Introduction

1. The starting (first two lines) of the introduction section seems complex for the reader. Starting with a simple sentence would be better.

2. Twitter, with over 315 million …., here need to add a reference about the statistics.

Literature Review

3. Add some paper mentioned below will rich this section

A) Jang, H., Rempel, E., Roth, D., Carenini, G., & Janjua, N. Z. (2021). Tracking COVID-19 discourse on twitter in North America: Infodemiology study using topic modeling and aspect-based sentiment analysis. Journal of medical Internet research, 23(2), e25431.

B) Ahmed, M. S., Aurpa, T. T., & Anwar, M. M. (2021). Detecting sentiment dynamics and clusters of Twitter users for trending topics in COVID-19 pandemic. PloS one, 16(8), e0253300.

C) Boon-Itt, S., & Skunkan, Y. (2020). Public perception of the COVID-19 pandemic on Twitter: Sentiment analysis and topic modeling study. JMIR Public Health and Surveillance, 6(4), e21978.

Research Design and Methods

4. In this section, several open-source libraries are used in different parts. Put the footnote with an open source link (Github or other repository platforms)

5. Need to mention, what methods or procedures are used in normalization and lemmatization with appropriate references.

6. Need to add relevant citation where mentioned TF-IDF.

7. In the sequential LDA subsection, there are mentioned figures and tables. Need to visualize according to describe (if you describe figure 3, then your figure 3 comes first, then other tables or figures)

Overall

8. There are some grammatical errors in the manuscript, need to consider in the upcoming revised version.

6. PLOS authors have the option to publish the peer review history of their article (what does this mean?). If published, this will include your full peer review and any attached files.

Reviewer #1: No

Reviewer #2: **Yes: **Md Shoaib Ahmed

---

## [Author Response · Author response to Decision Letter 0]

16 Mar 2022

Editor and Reviewer Comments:

Editor

Thank you for submitting your manuscript to PLOS ONE. After careful consideration, we feel that it has merit but does not fully meet PLOS ONE’s publication criteria as it currently stands. Therefore, we invite you to submit a revised version of the manuscript that addresses the points raised during the review process.

Additional Editor Comments (if provided):

Both reviewers think the manuscript has a merit for publication; however, they also think it needs more wok. Please read the comments and revise accordingly.

Thank you very much for reviewing our manuscript and providing constructive comments! We have revised the manuscript according to editor and reviewers’ comments to the best of our knowledge.

Journal Requirements:

and

Thank you for providing the PLOS ONE’s style requirements. We have revised the manuscript according to the templates. We used the three levels heading, capitalizing only the first word. We also reformatted figures and tables citations and authors affiliations. 

2. In your Methods section, please ensure that you have included a statement specifying whether the data collection method complied with the Twitter terms of service.

This is a valid point! Per comment, we included a statement specifying the data collection method complied with the Twitter terms of service:

The data collection method complied with the Twitter terms of service. The Twitter rules and policies can be found here: https://twitter.com/en/tos

We removed the term “data from this study are available upon request”. Here’s an excerpt from Twitter’s Terms of Service: “You may only distribute Tweet IDs, Direct Message IDs, or User IDs”. This means that we’re not able to share datasets containing the content of tweets. We revised the statement in the cover letter:

According to Twitter’s Terms of Service, we may only distribute Tweet IDs, Direct Message IDs, or User IDs. This means that we’re not able to share datasets containing the content of tweets. We stripped the datasets and provided a list of Tweet IDs, please refer to the file: twitter_data_tweet_ids

“This work was sponsored by the Zaanheh Project and Center for Data Science and Artificial Intelligence at New York University (Shanghai); Fujian Urban Investment and Technology Institute’s Research Fund (Grant No. 20210201 FJCT).

“This work was sponsored by the Zaanheh Project and Center for Data Science and Artificial Intelligence at New York University (Shanghai) The funders had no role in study design, data collection and analysis, decision to publish, or preparation of the manuscript.”

We removed funding information from the Acknowledgements section of the manuscript. We revised the cover letter with the updated funding information:

Comments to the Author

Reviewer #1: 

The articles have several issues which I want to present here.

One main issue is that whether it is title, abstract, or research questions everywhere the authors have talked about public sentiments but no sentiment analysis has been performed in this research paper. This research paper is only presenting the results from topic modelling. This is very confusing. I don't understand why the authors is written about public sentiment when they are not performing sentiment analysis.

Thank you very much for reviewing our manuscript and providing constructive comments! We have revised the manuscript according to your suggestion. We removed public sentiment from the title. We also revised the abstract and research questions:

Dynamic topic modeling of Twitter data during the COVID-19 pandemic

 (1) What kinds of topics are individuals and groups vocalizing in relation to the pandemic? (2) Are there any noticeable topic trends and if so how do these topics change over time and in response to major events?

1. Introduction

Organization of paper can be more specific (rewrite it in a better way and as a separate paragraph)

This is a great point! We have rewritten the organization of paper to be more specific as a separate paragraph:

The rest of the paper is organized as follows: In the literature review section, we examine the contemporary literature relating to topic modeling as it relates to twitter data and epidemiology and recent developments in Predictive Analytics and Natural Language Processing. In the research design and methods section, we describe data acquisition, data ingestion and prepossessing, and Sequential Latent Dirichlet Allocation. In the results section, we delve further into topic distribution and interpreting topic representations, before finally discussing SeqLDA limitations on topic structure and explaining unpopular topics and over-generalization. The conclusion section states how we contributed to an understanding of both the topics surrounding the COVID-19 pandemic and their evolution over time. 

2. Literature Review

Literature Review is vague. It should be focused on the related works that include topic modeling, COVID-19, Social media (Twitter).

We agree with the reviewer. We have revised the literature review focusing on topic modeling, COVID-19 and Twitter. We renamed and combined the original sections 2.2 and 2.3. We also added the following paragraphs to the manuscript:

The outbreak of the Novel Coronavirus Diseases (COVID-19) has spread across the globe since late 2019. It has caused significant impacts on people's daily life and taken hundreds of thousands of lives away (Yin et al., 2021). The lockdown and vaccination policies also affected billions of people and the impacts on global economy, transit, and public health are profound (Liu et al., 2020; Yao et al., 2021). According to WHO, as of February 2022, there are over 430 million confirmed cases, 5.9 million confirmed deaths, and more than 10 billion vaccine doses administered (WHO, 2022). Nonetheless, this is not the first-time mankind is facing a pandemic... 

The ongoing pandemic of COVID-19, especially the Delta variant and Omicron variant, prompted urgent needs for more in-depth studies of COVID-19 and Twitter in predictive analytics. For example, in a study by Rajput et al. (2020), Tweets posted by both social media and WHO were investigated. They found more positive responses to COVID-19 than negative emotions. Other studies explored the associations between COVID-19 and the human mobility restrictions, lockdown, and social distancing and on limiting the spread of the virus (Block et al., 2020; Kraemer et al., 2020; Lau et al., 2020; Li et al., 2021; Li, Peng, He et al., 2021). More recently, Twitter data has been leveraged to help understand the utility of public sentiment and concerned topics in public health. Early work in 2020 by Boon-Itt and Skunkan (2020) implemented topic models against a small subset of tweets spanning the first months of the pandemic, identifying multiple stages of public awareness as the virus spread, as well as different vocabularies associated with positive and negative sentiments on the outbreak. Jang et al. (2021) examining the relationship between public health promotions and interventions and public perceptions in Canada, leveraging an aspect-based sentiment analysis (ABSA) model to generate topics and sentiment. Utilizing a novel LDA approach, Ahmed et al. (2021) captured user-specific sentiment and sentimental clusters over time and ranked users according to their activity in trending topics, in an effort to understand user behaviors amidst varying pandemic topics.

This line - As of June 2019, Twitter has made geo-tagging an opt-in feature (only 1-2% of tweets are now geo-tagged) - is not clear.

Thank you for pointing this out! We have rephrased this to clarify that opt-in means users must actively request Twitter to add location to their tweets, reducing the volume of tweets that have this data available:

As of June 2019, Twitter has made geo-tagging an opt-in feature, meaning users must actively request Twitter include their locations in Tweets (only 1-2% of tweets are now geo-tagged as a result).

Section 2.4 Contribution of this work

This have a spelling mistake in Spanish Flu which is written as Spanish Flue. Also, novelty of the work can be explained in better way (improvement needed)

We have corrected the misspelling. Also, the novelty of the work has been rephrased to emphasize its scale, open-source nature, and analytical contributions. We added references to our publicly accessible GitHub repository as a footnote:

Previous works reviewed on the topic of the COVID-19 pandemic have been successful in identifying relevant topics for public health purposes, but have limited themselves to smaller datasets over broader periods of time. This work has focused on delivering an end-to-end scalable Topic Modeling Pipeline, with a publicly accessible GitHub repository outlining methods and technologies used, successfully achieving a scale of millions of analyzing millions of tweets per day. Additionally, our work leverages dynamic versions of LDA to measure topic drift on a topic and vernacular level, helping to identify changes in trending topics at scale and dynamically over time. 

2 https://github.com/akbog/urban_data

3. Research Design and Methods

Section 3.2 Data Ingestion and Preprocessing

It is not clear whether data in this Line - The dataset was scraped over a 14-day period starting March 31st and is composed of over 46 million tweets, averaging over 3 million tweets per day. - is before preprocessing or after preprocessing.

Preprocessing. We have revised the manuscript:

The dataset was scraped over a 14-day period starting March 31st and is composed of over 46 million tweets, averaging over 3 million tweets per day, before preprocessing.

4. Results

Section 4.1 Topic Distributions

Second paragraph is not incomprehensible.

Per comment, we have revised the second paragraph of section 4.1:

We also observe an increasing trend in the representation of the top 12 topics over the specified period. A few potential explanations could be: (1) As the Sequential Model trained over each successive time-slice, those topics making up the bulk of our dataset were well represented, note the steep increase on the first day of training; (2) There exists a strong correlation between the change in size of our day-to-day training population and the representation of the top 12 topics.

Section 4.2 Interpreting Topic Representations

Table 4 shows ten topics which I guess are the biggest ten topics but how can we know this? Topic weightage, or topic size usually shows this but this research paper is not showing any of this. I suggest the authors to show in table 4 topic size also.

How (on what basis/criteria) the 5 topics in Figure 7 is selected?

We have added a clarifying sentence to describe that the topics in Table 4 are the 10 largest, representing over 70% of our dataset at any point in time. We have also added a Topic Size column to our table and reordered the table in order of topic size.

Topic Interpretations 

Topic # Words Label Topic Size

46 time, like, need, know, world, day, life, think, going, good Status Updates 52%

55 trump, president, american, america, democrat, vote, response, china, republican, obama US Politics 5%

23 death, test, number, testing, case, vaccine, infection, rate, data, patient Infection & Testing 5%

52 case, death, new, state, new_york, total, update, county, city, reported Reports 4%

1 online, business, help, student, support, resource, free, pro- gram, new, school Personal Finances 2%

20 stay_home, stay_safe, social_distancing, safe, stay, home, lockdown, save_lives, healthy, easter Social Distancing 1.5%

17 mask, worker, nurse, ppe, hospital, patient, medical, front- line, face_mask, healthcare_worker Healthcare 1.5%

39 trump, state, january, response, election, economic, warned, american, government, warning American Response 1.5%

21 support, community, thank, help, crisis, health, response, team, excellent, time Positive Response 1%

35 supply, staff, ppe, company, equipment, worker, player, medical, employee, testing Medical Resources 1%

We provided better contextualization for the 5 topics selected in Figure 7, which provides a qualitative interpretation of topic drift over our study period:

Figure 7 summaries the topics outlined above (i.e. Easter, and Social Distancing) and demonstrates how our DTM manages to capture their relevance over time. Terms such as “student” rise in importance as Universities begin to settle into remote education, whereas terms such as “small_business” remain steady, as conversations continue to persist over the impact of COVID-19 on small businesses and the potential for stimulus.

5. Discussion

Section 5.1 SeqLDA Limitations on Topic Structure

Authors selected 30 topics but they did not provide any explanation about how they came up with this number?

We provided an explanation that the tradeoff in Coherence Score, which was used to determine the original model was no longer valid when the dilutive terms were removed from our corpus. It becomes clear from the visualizations that these terms were dividing topics that were really overlapping as well as merging topics that were in fact distinct.

We exclude words with dilutive properties and limit the number of topics to 30, as we found the tradeoff between overlapping topics did not warrant the small gain in Coherence score (see Figure. 4), once the list of dilutive terms had been excluded. The results clearly demonstrate the impact that these groups of words, as well as the number of topics, has on the structure of our document groups. It’s clear that the new visualization is able to cluster 30 topics into distinguishable non-overlapping groups without these terms, and this strengthens the original intuition of topic 46, which encompassed greater divisions of our topic space thanks to the general usage of terms such as “time”, “need”, or “day”.

6. Conclusion is not well written.

Lastly, authors have written that the data of this research paper is available at NYU Shanghai Library but they provided nor access number or link to access this data. They should make all the data available so that reviewers can access it and verify this research's authenticity.

Thank you for providing constructive comments! We have provided the data set via this link: Per Twitter’s Data Sharing Policies, we can make available only the Twitter IDs from our research, which we are currently hosting in this file: twitter_data_tweet_ids 

This version of our dataset is filtered from the full 46 million tweets scraped (which were not retained) to include 12,575,691 English Tweet IDs, which were the baseline of our study, prior to pre-processing. We also revised the manuscript:

Our approach differentiated itself in both scale and scope, utilizing advanced SeqLDA to study the emerging topics concerning COVID-19 at a scale that few works have been able to achieve, but that many will be able to reproduce, given the open source and architecture heavy nature of our research.

Reviewer #2: 

Introduction

1. The starting (first two lines) of the introduction section seems complex for the reader. Starting with a simple sentence would be better.

Thank you very much for reviewing our manuscript and providing constructive comments! We have revised the first few lines into simple sentences:

The last two decades have seen societies continue to evolve their means of virtual socializing and self-expression. Consequently, simultaneous advancements in the primary statistical domains related to communication via social media (i.e. Natural Language Processing (NLP)) are observed (Conway et al., 2019; Hirschberg and Manning, 2015; Farzindar and Inkpen, 2017).

2. Twitter, with over 315 million …., here need to add a reference about the statistics.

Per comment, references are provided. We also updated the statistics according to the latest Twitter Corps Shareholder report:

Twitter, with over 199 million monetizable daily active users (mDAU) generating over 500 million tweets per day as of Q1 2021, has historically served as a reliable source of social expression, largely as tweets tend to contain the following useful properties: textual data (topics), temporal data (time-series component), and spatial data (geo-tagging and profiles) (Twitter Corp. 2021).

Literature Review

3. Add some paper mentioned below will rich this section

A) Jang, H., Rempel, E., Roth, D., Carenini, G., & Janjua, N. Z. (2021). Tracking COVID-19 discourse on twitter in North America: Infodemiology study using topic modeling and aspect-based sentiment analysis. Journal of medical Internet research, 23(2), e25431.

B) Ahmed, M. S., Aurpa, T. T., & Anwar, M. M. (2021). Detecting sentiment dynamics and clusters of Twitter users for trending topics in COVID-19 pandemic. PloS one, 16(8), e0253300.

C) Boon-Itt, S., & Skunkan, Y. (2020). Public perception of the COVID-19 pandemic on Twitter: Sentiment analysis and topic modeling study. JMIR Public Health and Surveillance, 6(4), e21978.

Thank you for providing the three papers. They are highly related to our paper and we have incorporated all three into the literature review section:

More recently, Twitter data has been leveraged to help understand the utility of public sentiment and concerned topics in public health. Early work in 2020 by Boon-Itt and Skunkan (2020) implemented topic models against a small subset of tweets spanning the first months of the pandemic, identifying multiple stages of public awareness as the virus spread, as well as different vocabularies associated with positive and negative sentiments on the outbreak. Jang et al. (2021) examining the relationship between public health promotions and interventions and public perceptions in Canada, leveraging an aspect-based sentiment analysis (ABSA) model to generate topics and sentiment. Utilizing a novel LDA approach, Ahmed et al. (2021) captured user-specific sentiment and sentimental clusters over time and ranked users according to their activity in trending topics, in an effort to understand user behaviors amidst varying pandemic topics.

Research Design and Methods

4. In this section, several open-source libraries are used in different parts. Put the footnote with an open source link (Github or other repository platforms)

Per suggestion, we have added footnotes with open source links in GitHub:

https://github.com/nltk/nltk

https://github.com/RaRe-Technologies/gensim

https://github.com/apache/spark

https://github.com/JohnSnowLabs/spark-nlp

5. Need to mention, what methods or procedures are used in normalization and lemmatization with appropriate references.

We provided references to SparkNLP’s Normalizer, Lemmatizer, and Stemmer Annotators used in the analysis:

Additionally, we normalize (i.e. standardize case and remove punctuation) and lemmatize (i.e. revert to common base word) each tweet utilizing SparkNLP’s Normalizer, Lemmatizer, and Stemmer annotator classes to produce a standardized vocabulary.

https://nlp.johnsnowlabs.com/docs/en/annotators

6. Need to add relevant citation where mentioned TF-IDF.

Relevant citations on the origin of TF-IDF have been added and an explanation of its usage:

… Inverse Document Frequency (TF-IDF), term weighting strategy developed in the early 1970s and still used in the majority of NLP applications today (Jones, 1972), …

7. In the sequential LDA subsection, there are mentioned figures and tables. Need to visualize according to describe (if you describe figure 3, then your figure 3 comes first, then other tables or figures)

Thank you for pointing this out! We moved Figure 3 to the front.

Overall

8. There are some grammatical errors in the manuscript, need to consider in the upcoming revised version.

We conducted language editing for grammatical errors.

---

## [Decision Letter · Decision Letter 1]

4 May 2022

Dynamic topic modeling of Twitter data during the COVID-19 pandemic

PONE-D-21-27183R1

Dear Dr. Guan,

We’re pleased to inform you that your manuscript has been judged scientifically suitable for publication and will be formally accepted for publication once it meets all outstanding technical requirements.

Kind regards,

Kazutoshi Sasahara

Academic Editor

PLOS ONE

Additional Editor Comments (optional):

On behalf of reviewer 2, I checked the minor comments have been properly addressed, and thefore made this decision.

Reviewers' comments:

Reviewer's Responses to Questions

**Comments to the Author**

1. If the authors have adequately addressed your comments raised in a previous round of review and you feel that this manuscript is now acceptable for publication, you may indicate that here to bypass the “Comments to the Author” section, enter your conflict of interest statement in the “Confidential to Editor” section, and submit your "Accept" recommendation.

Reviewer #1: All comments have been addressed

2. Is the manuscript technically sound, and do the data support the conclusions?

Reviewer #1: (No Response)

3. Has the statistical analysis been performed appropriately and rigorously? 

Reviewer #1: Yes

4. Have the authors made all data underlying the findings in their manuscript fully available?

Reviewer #1: Yes

5. Is the manuscript presented in an intelligible fashion and written in standard English?

Reviewer #1: Yes

6. Review Comments to the Author

Reviewer #1: Authors have updated the manuscript according to the comments provided earlier. The manuscript has improved and is publishable.

7. PLOS authors have the option to publish the peer review history of their article (what does this mean?). If published, this will include your full peer review and any attached files.

Reviewer #1: No

---

## [Editor Report · Acceptance letter]

16 May 2022

PONE-D-21-27183R1 

Dynamic topic modeling of Twitter data during the COVID-19 pandemic 

Dear Dr. Guan:

I'm pleased to inform you that your manuscript has been deemed suitable for publication in PLOS ONE. Congratulations! Your manuscript is now with our production department. 

Kind regards, 

on behalf of

Dr. Kazutoshi Sasahara 

Academic Editor

PLOS ONE